# COVID-19 Pandemic Lockdown and Wellbeing: Experiences from Aotearoa New Zealand in 2020

**DOI:** 10.3390/ijerph19042269

**Published:** 2022-02-17

**Authors:** Tara N. Officer, Fiona Imlach, Eileen McKinlay, Jonathan Kennedy, Megan Pledger, Lynne Russell, Marianna Churchward, Jacqueline Cumming, Karen McBride-Henry

**Affiliations:** 1Te Hikuwai Rangahau Hauora|Health Services Research Centre, Te Herenga Waka—Victoria University of Wellington, Wellington 6140, New Zealand; hsrc@vuw.ac.nz (F.I.); megan.pledger@vuw.ac.nz (M.P.); lynne.russell@vuw.ac.nz (L.R.); marianna.churchward@vuw.ac.nz (M.C.); jackie.cumming@vuw.ac.nz (J.C.); 2Department of Primary Health Care and General Practice, University of Otago (Wellington), Wellington 6021, New Zealand; eileen.mckinlay@otago.ac.nz (E.M.); jonathan.kennedy@otago.ac.nz (J.K.); 3School of Nursing, Midwifery and Health Practice, Te Herenga Waka—Victoria University of Wellington, Wellington 6140, New Zealand; karen.mcbride-henry@vuw.ac.nz

**Keywords:** COVID-19, wellbeing, Aotearoa New Zealand, mental health, social distancing, lockdown

## Abstract

In 2020, in the first COVID-19 pandemic lockdown, Aotearoa New Zealand consistently maintained stringent public health measures including stay-at-home lockdowns and distancing responses. Considering the widespread disruption to social functioning caused by the pandemic, this paper aimed to explore environmental and social factors that influenced the wellbeing of individuals during the first lockdown in Aotearoa New Zealand. Our mixed-methods study involved a survey (n = 1010) and semi-structured interviews of a subset of surveyed individuals undertaken at the tail end of the first 2020 lockdown. Survey participants were recruited through social media-driven snowball sampling, less than 50% were aged under 45 years and 85% identified as female. Of those interviewed, 63% identified as female. Qualitative interview findings and open-ended survey results were analysed thematically. Participants described a variety of factors influencing wellbeing, largely related to the community and household; physical, behavioural, and lifestyle factors; access to health services; and social and economic foundations. While much of the focus of COVID-19 recovery was on reversing the economic and physical toll of the pandemic, our findings emphasise the need to empower individuals, families, and communities to mitigate the pandemic’s negative implications on wellbeing.

## 1. Introduction

In 2020, the World Health Organization expressed concerns around the impact of COVID-19 on mental health and psychological wellbeing in populations including essential health care workers, older adults, carers of children, those who are disabled, and people in isolation [1,2]. Since then, there has been increasing concern for the psychological and social impacts of the pandemic on different populations [3,4,5]. For example, in the United Kingdom, mental health deteriorated during the pandemic [6], particularly for those with pre-pandemic experience of mental distress, with risk factors noted as social isolation, job and financial losses, housing insecurity and quality, working on the frontline, loss of coping mechanisms, and reduced access to mental health services [7,8,9,10]. In other United Kingdom research, better perceived mental health at the beginning of lockdown was associated with the perception of increased kindness, community connectedness, and being an essential worker [11]. Australian research using a nationally representative sample of adults conducted early in the pandemic found higher levels of anxiety and depression owing to social, work, and financial disruptions [12]. A 2020 international survey of mental health during the pandemic found the highest burden of mental health difficulty in 10% of the population, with around 40% of the study population coping well [5]. Other 2020/21 longitudinal studies in countries including the United Kingdom, China, Ireland, and Italy also suggest variable changes in psychological outcomes throughout the pandemic [10,13,14,15,16]. These studies show the varied impact of COVID-19 on mental health worldwide.

The COVID-19 pandemic, the resultant restrictions, and stay-at-home lockdown measures have multidimensional social consequences, with the potential to exacerbate existing disparities. The pandemic can directly affect mental health, through fear, concern, loss, and grief, and COVID-19 can directly cause neuropsychiatric changes in infected people [17]. Rumination and emotional suppression are known psychological mechanisms that predict poorer mental health in lockdown [18]. Other effects are indirect due to the impact of lockdowns on isolation, loss of freedom, immediate or longer-term economic/financial repercussions, changes in health-related behaviours, and food insecurities [19,20]. In addition, increasing loneliness may occur [21,22,23], although some studies early in the COVID-19 lockdowns showed little impact [24,25].

Initial pandemic research hypothesised that adverse changes to resilience and child-family wellbeing were likely [26]. More recent research suggested high anxiety and burnout amongst parents and increasing negative and positive expressed emotions [27,28,29], although, parenting flexibility appeared linked to greater family cohesion [30]. For women (and somewhat for parents), the pandemic has been described as exacerbating a triple work burden related to increasing productive, family, and community obligations [31,32,33], although not all findings support this [34]. Research from Ireland also highlighted the mixed relationship between perceived wellbeing and activities of daily living in a socially distanced and locked down environment. The study suggested that outdoor activities, such as exercise or gardening, may mitigate negative feelings, while home-schooling, using social media, or listening to COVID-19-related news exacerbated negative feelings [35].

### 1.1. Aotearoa New Zealand COVID-19 Response

On 28 February 2020, the first case of COVID-19 was confirmed in Aotearoa New Zealand. Following this, Prime Minister Jacinda Ardern announced the closure of the border and declared a national state of emergency on 25 March as part of a ‘go hard, go early’ policy to prevent COVID-19 transmission. The country moved into a stringent stay-at-home nationwide lockdown, lasting 33 days, which effectively prevented virus transmission and meant that the country had comparatively very low confirmed and probable cases and consequently few deaths from COVID-19 [36,37,38]. Notably, however, the country’s Indigenous Māori and Pacific populations remained at greater likelihood of COVID-19 hospitalisation and had an estimated increase in fatal infections [39,40,41]. Aotearoa New Zealand’s four-level COVID-19 alert system, with level four stay-at-home lockdown, was considered amongst the most stringent response to the pandemic globally [42]. At level four, schools and other non-essential businesses remained closed and those working in these businesses were required to stay at home and work from a distance within their own “bubble” (i.e., a safe family household that may pop if contact is made with those outside it) [43]. These bubbles typically contained one household, with approximately 70% housing an essential worker or vulnerable person [44]. The country and individual regions moved in and out of alert levels during 2020 and 2021 due to COVID-19 being introduced into the community from returnees in Managed Isolation Quarantine (MIQ) facilities. With high vaccination rates in late 2021 (94% of all eligible individuals double-vaccinated in January 2022, although notably lower vaccination rates in Māori (84%) [45]) the government introduced a new traffic light distancing system that unforeseen, coincided with the introduction of the Omicron variant into the country via MIQ [46].

#### Implications for Mental Health and Wellbeing

While the stringent lockdowns led to prolonged success in eliminating COVID-19 during 2020, this came with a psychological toll [47,48,49], particularly amongst those with pre-existing mental health conditions [48,50], youth [51] and younger adults [50], and essential workers [52]. This psychological toll was seen in increased overdose and self-harm hospital presentations and significant increases in ambulance callouts nationally for mental health conditions over lockdown [53,54]. People experienced financial stress and housing problems [55,56], high initial anxiety about getting COVID-19 and loneliness/isolation [57], with job and income loss associated with poorer wellbeing [58]. Māori, Pacific people, and refugees also experienced further barriers to wellbeing owing to structural and communication inequities exacerbating housing and economic issues [56]. However, those living in Aotearoa New Zealand described both increased or decreased alcohol consumption and smoking during lockdown, and many people also recognised positive aspects from lockdown [51,59,60]. Such positives included more time for hobbies and family resilience, particularly amongst Māori [51].

Recognising the potential psychosocial impact of the pandemic on Aotearoa New Zealand, the Prime Minister encouraged the population to “be kind”, and the Government launched a recovery framework [61,62] focussed on five areas (Table 1). This framework intended to create national and local resource alignment and support individuals, whānau (family), and communities to respond to the impact of COVID-19. As of April 2021, Aotearoa New Zealand was the only country with a dedicated mental health recovery plan [63].

### 1.2. Research Focus

The present research explores the impact of the COVID-19 pandemic and the first 2020 lockdown on people’s wellbeing in Aotearoa New Zealand. While there is considerable literature quantifying the existence of mental health and wellbeing changes over 2020 lockdowns, there is limited literature exploring individual perspectives on these wellbeing changes. We elicited viewpoints on the individual, whānau, community, workplace, and health service factors that acted to enhance or diminish the lockdown experience. Understanding the impact of these factors is pertinent in a global environment dominated by the (long-term) impact of lockdowns.

## 2. Materials and Methods

Applying a mixed-methods approach, adults (aged 18 years or older) who either had or wanted contact with health services during the first national 2020 lockdown participated in an anonymous online survey (Figure 1). Of the 1010 survey participants, 436 supplied contact details for follow-up and 38 of these respondents were interviewed (see Appendix A for a copy of the interview schedule) via telephone or videoconferencing from 4 to 28 May 2020. This period meant that more than half the interviewees were interviewed during levels 3 and 4 (with stay-at-home measures in place), while the rest were interviewed in alert level 2 (ability to re-enter the community with physical distancing).

Interviewees gave informed consent for an audio/Zoom-recorded interview, could request to be interviewed by a Māori or Pacific interviewer, and could review their transcript for accuracy. Interviews focussed on participant experiences of health care during lockdown and how they managed their health. Most interviewees (35/38) spontaneously elaborated on responses to stress and isolation; where participants requested additional support, the research team offered information on health and wellbeing support lines. Interviewers offered to stop or pause their interviews if there was distress, but no one took up this option. Interviewees received a voucher as thanks for their time. Those who requested a copy of their transcripts were also able to review these; no one revised their transcript.

Lockdown restrictions meant that the survey and interview schedule were informally reviewed by external experts and pilot tested amongst the research team’s households. Interviewees were more commonly female and more likely than survey respondents to be older and not seeking employment (see Table 2). Participants were also commonly from central regions of Aotearoa New Zealand, likely due to the snowball recruitment method and the research team’s institutional base being centrally located.

### Qualitative Analysis

This paper focuses solely on a thematic analysis [64] of qualitative data from (1) the 38 interview transcripts and (2) responses to open-ended survey questions. Reporting of interview quotes from participants choosing to be interviewed by a Māori interviewer (n = 5) are not included in this publication because of our commitment to Indigenous data sovereignty; these will be separately analysed.

There were seven open-ended questions in the survey (excluding those in the demographic section); none specifically asked after mental health or wellbeing. Most responses on this topic came from Q56, ‘is there anything else you would like to tell us…?’ Responses to questions around what worked well or not about telehealth, in-person consults, and health in general were also included in the analysis if participants commented on their mental health or wellbeing.

Applying the COVID-19 psychosocial and mental wellbeing recovery framework [61,62] as an initial coding frame (Table 1), we developed distinct themes around protective resilience factors and areas causing mental distress within the framework’s five focus areas. In following the principles of thematic analysis, we were able to gain nuanced insights into each of the framework’s areas, without being bound by theoretical constraints, unlike other methodology approaches, such as grounded theory [64,65]. We also wanted to use an applied research approach so that findings could directly inform decision making.

Two members of the research team first analysed data using NVivo 12 (QSR International Pty Ltd., Melbourne, Australia) to manage the data. This initial analysis was iteratively reviewed, checked, and interpreted by two further team members. Findings were then discussed within our interdisciplinary research team to ensure agreement and so that major findings were not inadvertently omitted, in line with principles articulated by Braun and Clarke [64]. This team included clinicians, patient-experience researchers, and those providing specific cultural perspectives. Having such a varied team facilitated reflexivity, a principle inherent in Braun and Clarke’s approach [64,66]. Themes identified in interviews were checked against survey themes following their analysis, this ensured analysis alignment and comprehensiveness. Quotes used in the following section are taken from interview (I) or survey (S) participants and provided alongside information on the participant’s age range and gender.

## 3. Results

Within the five focus areas in Table 1, various subthemes became apparent throughout the analysis underpinned by social and economic foundations (Table 3). These subthemes relate to psychosocial factors affecting wellbeing (whānau and individuals) and interactions with others (households, neighbours and communities, and health providers). Notably, participants did not differentiate between primary and specialist mental health services; we have grouped these service delivery areas when presenting our findings. Where participants have used words in Te Reo Māori (the Māori language), these have also been translated.

### 3.1. Community-Led Solutions

Participants discussed the role of community support in empowering them to respond to mental distress. This support involved a balancing act wherein household relationships could bolster (or impede) wellbeing, and household responsibilities led to additional demands/strains on workplace ‘loyalty’. Additionally, participants commented on the role of telecommunications and neighbourly networks.

#### 3.1.1. Household Relationships and Responsibilities

Emotional support, connections to others, and relationships can mitigate mental distress, helping to reinforce a sense of identity, belonging, and reduced fear. During lockdown, some people expressed joy in having more quality time to spend with family members, particularly across generations, spending time with children and grandchildren who were in their bubble. Others commented on having additional support to manage the household. They saw this as a positive outcome of COVID-19.


*What a different experience it would have been had we been here on our own… Having the two littlies certainly was not an unpleasant experience because we had the constant company and the constant joyfulness of young people… It was quite a different experience to a lot of my colleagues and friends who found it very isolating and very lonely.*
(I: 65-74, F) 

However, lockdown could strain relationships by forcing people to live in uncomfortable proximity, especially when changes related to lockdown added more stresses (e.g., unemployment or loss of income).


*Since lockdown, I have my son staying with me while he awaits social housing. He suffers from schizophrenia which can be extremely stressful to live with in my small 1bdrm flat. He is on waiting list for a flat but nothing is happening due to lockdown so it is very hard. I have lost 10 kgs in weight.*
(S: 55-64, F)

When describing the effect of lockdown on relationships with partners, husbands, or wives, participants often focussed on negatives. They described wanting a change after being stuck with the same person, or looking forward to “sending the husband back to work… [as] he’s nearing being kicked the f*** out” (S: 25-34, F).


*To be honest it’s made me a little bit unhappy just being cooped up with one person.*
(I: 65-74, M)

Participants also described being able to spend more time together and to take care of each other. However, they acknowledged a dilemma where working under normal (business as usual) conditions may limit the ability to concurrently enjoy close family time. In turn, participants suggested that the burden of additional household responsibilities, particularly around overseeing the schooling of children and caring for older relatives, created tensions, especially for those who continued working. Participants described seeking calm while balancing competing demands between “wild” (S: 25-34, F) children, elderly parents, and work.


*Working from home with children has been extremely stressful. You are constantly having to prioritise your kids’ education or your work. Who are you loyal to? The one giving you money or the ones you gave life to?*
(S: 25-34, F)

On the other hand, participants acknowledged that family connection meant different things based on cultural values, reinforcing the need to consider how best to foster (extended) family unity, while maintaining appropriate distancing measures.


*Households don’t necessarily represent… family connections [for Pacific]. You have to allow that opportunity to… co-bubble with different households.*
(I: 45-54, F)

Allowing household bubbles to expand during Alert Level 3 to include close family and whānau, caregivers, or isolated people [43] was important in terms of maintaining connections and could help with sharing household and childcare responsibilities (Section 3.1.1).


*Because my mother-in-law lives alone, we discovered it was within the rules to invite her into our bubble… [It] was just a win-win situation… with her watching her grandson and getting some interaction… It gave both of us space to be able to do our work.*
(I: 25-34, M)

Recognising that a change in alert levels occurred because of an increased risk of community COVID-19 transmission, another participant highlighted that balancing responsibilities began before an official lockdown was announced.


*I have a parent who lives alone, an elderly parent, I am her person… As soon as the [first alert] level announcement happened… it was like “right, that’s it, you [elderly parent] are not leaving the house”… I have to do… [everything] she needed… [It was] quite a lot of pressure… if I fall over, there is nobody else.*
(I: 45-54, F)

#### 3.1.2. Telecommunications

A positive aspect of lockdown was that people adopted ways of engaging with others using telecommunications (especially videoconferencing applications, such as Zoom). Although a new skill for some, it resulted in connections that buffered against social isolation and replaced aspects of in-person contact. Participants reported increased family contact.


*My parents who are elderly in Dunedin and my siblings spread around the country, we set up a videoconferencing group, so we connect every three or four nights.*
(I: 55-64, F)

Interviewees described this contact as extending beyond their home bubble, to include people globally. Such connections facilitated socialisation, family check-ins, and an ability to discuss concerns around wellbeing.


*I started doing some Skype calls or WhatsApp calls to touch base with more family members during lockdown because I was needing that… I got back in touch with some friends and family that I didn’t for some time.*
(I: 25-34, F)

One participant highlighted the importance of telecommunications for migrants.


*With family overseas,… you need to calculate all the time the time difference… If I want to speak with my mum right now, I can’t call her because it’s 4 am… I just kept in touch a lot more… I really needed it in lockdown.*
(I: 25-34, F)

Participants highlighted that certain groups may be excluded from using telecommunication tools, including those with skill/knowledge barriers or those unable to afford the costs. Notably, within Aotearoa New Zealand, not all individuals have access to the internet or telephones [67]. In cases of lockdown, no telecommunications access can mean no connections.


*People who don’t know how to access the internet… We would have to be here to set it up for Dad, otherwise, he won’t be able to access his Zoom.*
(I: 45-54, F)

Additionally, participants also acknowledged the likelihood of maintaining ongoing digital connections once out of lockdown, perhaps suggesting a change in how families and friends support each other.


*I socialise with a lot of friends just through Zoom… I’ll probably keep that up,… twice a week I catch up with my family who are spread [globally].*
(I: 25-34, M)

#### 3.1.3. Community and Neighbourly Relationships

Alongside digital connections, some people discussed the role of neighbourly and community relationships to maintain their personal wellbeing. Much of this discussion focussed on access to groceries and food parcels, where participants described feeling “cared for” (S: 45-54, F).


*I’m thankful for the free food parcels that the local community organisations have been giving out, as some families have faced hardships… and it would have made a big impact on their health and wellbeing.*
(S: 45-54, F)

In contrast, others described a lack of neighbourliness, including situations where supermarket deliveries did not occur, perhaps creating unnecessary risks for vulnerable populations.


*Disappointed by lack of neighbourly support. Confusion over [the rule of not] going to the supermarket because of over 70 age. Supermarket not delivering goods to customers.*
(S: 75-84, F)

The lockdown also affected people’s ability to support neighbours in other ways, with one participant highlighting this as follows:


*My neighbour down the road has health issues and because of the [co-occurring] drought, they ran out of water and usually I would say, “Come over and grab some water.”… [Instead,] I said, “Oh, I don’t think it’s a good idea because of COVID”.*
(I: 35-44, F)

Neighbours were seen as a source of potential support and advice, including for mental health needs. One participant highlighted the effect lockdown and physical distancing had on restricting the ability to have neighbourly support in the face of potential domestic violence.


*When things got tense with my flatmate… I reached out to my neighbours by phone… She said to “file a Police report”… You’re trapped inside lockdown…, there were no options. Even though I could reach out to my neighbours, none of them could come.*
(I: 45-54, F)

### 3.2. Whānau (Family) and Individuals Look after Their Mental Wellbeing

Nurturing or neglecting one’s mental wellbeing was touched on by almost all interviewees and many survey participants. They described ministering to themselves through physical lifestyle and behaviour changes, re-evaluating their pace of life and mindset; however, some felt stuck rather than safe at home.

#### 3.2.1. Physical Lifestyle and Behaviour Changes

Many interviewees discussed lifestyle changes to deal with lockdown. These could be ‘maladaptive’, such as drinking more alcohol to “wind down” (I: 25-34, M) or eating too much. Alternatively, they could be beneficial, such as increasing physical exercise and other activities (e.g., gardening, cooking, reading).

Changes in alcohol consumption at home varied and seemed to depend on previous patterns of alcohol intake and the support (or otherwise) of household members. Those who described themselves as drinking mostly in social circumstances tended to drink less, whereas others who drank to relieve tension or reward themselves drank more.


*I’m doing more AFDs [alcohol-free days] than I used to… It would be terribly easy to get into soaking up a bottle a night… I’m… mindful and don’t let it get away.*
(I: 55-64, M)

Some participants highlighted fluctuations in their ability to maintain physical behaviour change, such changes were framed in terms of the inability to continue good habits started before lockdown while managing feelings of “existential dread” (I: 65-74, F) or dealing with a temporary disruption to normal routines.


*I started out exercising more and I ended up drinking more… A couple of weeks ago I was like… ‘This is temporary, it’s fine if I just do whatever.’ And then I was thinking, ‘Maybe I should get back into good habits.’*
(I: 25-34, F)

Food consumption increased for some who reported weight gain. Others reported better quality food, with no takeaways, more home-cooked meals, and baked treat food as consolation. Participants discussed the influence of competing household responsibilities on physical lifestyle change and the struggle to maintain wellbeing.


*The big thing has been about creating routine,… having purposeful work and at times when my work has not felt purposeful, that’s some of the times I’ve really struggled.*
(I: 45-54, F)

During alert levels 3 and 4, exercise was a key strategy to manage stress, provide a “time out” (I: 45-54, F) from everyday triggers, a manageable goal, and a chance to get “fresh air” (I: 55-64, F). Participants described early morning walks to start the day and hopes to maintain good exercise outside of lockdown. Others described the inability to exercise because of changes in routines or lack of access to gyms.


*I am a diabetic, and my 10 h/day physical job is a large part of keeping my sugar levels in check. Being off-work, gyms closed, pools closed, not being allowed to kick a ball around…, I have regressed mentally to the point of being totally locked-in.*
(S: 55-64, M)

On the other hand, participants also commented that the slower pace of life (Section 3.2.2) meant added time for self-care and exercise, while still completing household and employment tasks.


*When we were in Level 4 and my husband was home, we were walking during the day, it was beautiful, and just doing things we never usually do… Because I’ve got a mobile phone for work and everything, I just took it with me. That kind of lifestyle freedom was- it’s beautiful.*
(I: 45-54, F)

#### 3.2.2. Pace of Life and Mindset

Related to the theme of behaviours/lifestyle change, was the shift in pace of life people experienced. This led individuals to reflect on how life was before COVID-19, and what positive aspects of lockdown life they wanted to hold on to. These included living a simpler life, appreciating local community connections, considering the environmental impacts of modern life, finding alternatives to ‘going out’ and commercialisation, and being positive and grateful for what they have.


*People were saying they couldn’t wait to get into shopping centres… You know when you’re in a shop,… you see things and you buy them, and I’ve found [without the shops] financially I feel better off,… It has been great, almost like a reset…*
(I: 45-54, F)

In turn, this also allowed people to reflect on experiences they had coming into lockdown:


*I was diagnosed with breast cancer just right at the start, so I actually think it’s worked really well because the whole world has slowed down. So, for me, being in lockdown actually meant that it wasn’t so stressful.*
(I: 45-54, F)

Others reflected that certain groups—essential workers—lost out on the ability to reflect, reset, and have reduced life pressure. One essential worker described this experience as follows:


*Everybody else was out going for walks with their dogs and biking and things. I left home when it was dark and come home when it was dark… My lifestyle over this whole period is worse… I had all these great intentions… but I just haven’t…*
(I: 45-54, F)

Isolation caused by lockdown was not necessarily perceived negatively. Those describing themselves as introverted tended to enjoy lockdown, having more quiet time, solitude, less exposure to people, and fewer issues to navigate.


*Almost 21 years I have napped… since we’ve been in lockdown I don’t need to sleep during the day… It tells me that being around so many people actually is exhausting.*
(I: 45-54, F)

In contrast, others described missing social contact and feeling disconnected from humanity. They expressed feeling pressure to make the most of lockdown and to “take action against things that have been difficult” (I: 45-54, F). Participants described this as potentially confronting.


*If we could just get through this blasted thing, with our mental health not too badly damaged, that’s a win. You’ve probably seen various comments on this online—if you don’t get through without learning three new skills, what kind of a person are you?... I’m a little bit careful not to set myself too ambitious goals for getting through this.*
(I: 55-64, M)

Spiritually, in exploring this idea of wider connections, some participants commented on how lockdown influenced their views on societal connections.


*I just have a greater appreciation for the sense of connection and reliance that we have on one another… I am reliant on everybody in the community and they are reliant on me. So, when I buy my coffee now, I’m consciously choosing who I buy it from and it’s an investment in their lives… that carries a different spiritual experience.*
(I: 45-54, F)

#### 3.2.3. Stuck or Safe at Home

The feeling of being stuck at home was difficult for some people. Participants used phrases such as “confined” (S: 75-84, F) and “like a prisoner in my own home” (S: 35-44, F) to describe their experiences. Those who liked less social contact or individuals accustomed to restricted or limited movements, such as those with health issues, fared better.


*Since I don’t go out and socialise,… it wasn’t really a big change. I’m actually finding now that I really like it, so the whole stuck at home syndrome or whatever they’re calling it now, I’m actually quite liking it.*
(I: 55-64, F)

In contrast were those who identified a sense of frustration and sadness and a need to keep busy in the face of limited interactions. The following participant, an individual living with two essential workers, explains:


*When D and R go to work [at essential jobs] and it just being me, eight hours a day and I would sort of be looking at these walls, and finding anything possible that I could clean… I am an extrovert, I need to be connecting with people all the time, that [lockdown] just became a bit hard.*
(I: 45-54, F)

Participants described the balance of maintaining physical safety through distancing versus looking after their own mental health. They rationalised the lockdown process as necessary and being “better than dying from [COVID-19]” (I: 75+, M) but as also limiting access to a normal life.


*I didn’t like getting stuck in the house… [But] if you look at the situation and weigh up everything, we are very lucky. Although there is a lockdown, we are still able to go out and go to the park… [But] you have to be doing something, you can’t wait in the house, that is waiting to die.*
(I: 75+, M)

One aspect of normal life affected by lockdown was the management of other health-related issues. For example, one participant was unable to visit their dying relative, another described needing to manage addiction issues, while many focussed on their existing mental health needs. Participants described these as immediate issues of concern, which caused more pressure for them and their whānau (family).


*I have suffered less from my arthritis… whilst working from home, and not having to commute a long distance by car, [but]… I have missed out on the flu vaccine I would normally have received… I have also had a regular mole map screening postponed, and a blood test and CV risk review have been postponed.*
(S: 55-64, F)

Fear of infection was strong, with many feeling safer at home. One survey participant explained how they chose not to visit health professionals because of concern around breaking their bubble. Others described fear of COVID-19 exposure as being more pronounced when considering at-risk populations, including parental populations, older people, those living with a disability or other chronic/high health needs.


*Both my parents are almost 80 and my ex-partner is a diabetic and he’s recovering from surgery and so I was like, right, well you three are incredibly vulnerable… I didn’t feel like I could play fast and loose with any of the restrictions.*
(I: 35-44, F)

Similarly, those who had to go out into the potentially COVID-19 infected community to work experienced more stress because of this fear, and the risk of contaminating home sanctuaries.


*[Going out] I had got quite distressed because the general energy… was an unpleasant sort of racy, kind of everyone on edge, super-suspicious kind of sketchy feel to it.*
(I: 45-54, F)

### 3.3. Primary and Specialist Mental Health and Wellbeing Support

Participants touched on the role of health system engagement in influencing wellbeing. They identified three main issues: service accessibility, the form of service delivery, and the clinician–patient relationship.

#### 3.3.1. Accessibility

Access to mental health services emerged as a key lockdown issue, participants described service provision as ranging from timely and proactive, to delayed and restricted. This variation in experience is impossible to explain without additional data. However, participants highlighted issues with finding information on how to access mental health support, restrictive admission, inpatient, and respite practices, and issues with availability of mental health helplines or substitute mental health professionals should there be widespread infection. The following participant described their experience with accessing national mental health helplines.


*My mother rang the 1737 line [Mental Health Helpline]… It took two phone conversations and 25 minutes of being on the line before anyone answered her. I understand that there is pressure on this sector due to not having enough service available… How many of our people have… ended their life because they haven’t been able to talk to someone.*
(S: 45-54, F)

Not everyone was clear about what mental health services were available or where they would go to get help in a crisis.


*I wouldn’t know how to access counselling services or mental health services in my community. GP, yes, but anything else, I don’t know,… If I was experiencing isolation, issues of suicide and I was on my own, I wouldn’t know, who do you ring, do you still ring Healthline because no one is on the other line of those crisis centres, are they still open?*
(I: 45-54, F)

One respondent felt unable to seek help from services perceived to be under strain, even after attempting suicide (notably, suicides reduced over lockdown [68]).


*I freaking attempted suicide… I felt I couldn’t even seek mental health care because the services were already stretched more and they had to go slower to disinfect everything between people. So, I got no help. Nobody knows what happened and I’m stuck still.*
(S: 18-24, F)

Participants also recognised the potential effect of COVID-19 on individual clinicians, understanding that the clinician they have a relationship with may get unwell or become unavailable. Similar concerns were expressed around the availability of medicines. These concerns affected wellbeing.


*I have been unable to access mental health support during the second half of the lock-down as my psychologist became unavailable. This has severely impacted my mental health during transition periods between alert levels.*
(S: 18-24, F)

Participants suggested that mental health needs were formally neglected over the lockdown, the following participant described their experience of accessing mental health respite care.


*The mental health service delivery for my husband and myself has been awful, with restricted access to respite even when beds are available, overly coercive practices during admission and unnecessarily restrictive processes within the unit that mean even patients in the open ward cannot go for a single walk outside each day escorted by staff.*
(S: 35-44, F)

#### 3.3.2. The Form of Service Delivery

The substantial COVID-19 transmission risk meant that hospitals and care facilities implemented restrictions on visitor numbers for in-patients and that much primary and specialist care was delivered from a distance. For some participants, seeing a health professional represented the first time they had physical contact during lockdown. One participant highlighted the surreal nature of this appointment.


*This is four weeks since somebody touched me, and it’s to draw my blood, it was just the weirdest thing… I lost something in the four weeks… I don’t know why it’s so meaningful but there is… a sadness,… grief.*
(I: 25-34, F)

Participants with experience receiving services in hospital over this time described distress and increasing feelings of isolation. Those with loved ones in hospital described negative impacts on their ability to support loved ones, including not being allowed to send gifts. Aotearoa New Zealand’s strict lockdown meant that people were generally unable to visit ill or dying relatives, even when these relatives were in their own homes. This was a lockdown-specific stress that had no mitigation.


*I suffer from anxiety and depression anyway, but unfortunately exacerbated a lot during lockdown because on top of that my dad’s got terminal cancer, and I was denied compassionate grounds to travel to see him by the Government. So that deeply affected me.*
(I: 45-54, F)

Several individuals discussed accessing mental health care via telehealth (video or telephone) with marked variation in preferences for support. For some, mental health issues could be readily managed by telehealth and could be an advantage when people did not want to leave the house (because of anxiety or depression—COVID-19 related, or just anytime), these individuals described telephone or video contact as able to get them through difficulties.


*Sometimes when it comes to asking the straight-up questions a GP asks about mental health conditions, it’s just a bit easier to do it over the phone rather than have a pair of eyes staring at you.*
(S: 18-24, F)

For others, telehealth caused added stress and anxiety, one participant went so far as to suggest additional risks may be posed to clinicians assessing mental health over the telephone. Others pointed out limitations in the ability of clinicians to assess physical cues when delivering services over the phone.


*If I were in a serious mental health crisis and I was to find myself on the phone with them; I’d find it really unsafe for them as well as me… There’s so much more they can assess… if I was physically present.*
(I: 25-34, M)

Some pointed out their discomfort with accessing telehealth care at home citing privacy concerns and distress in recounting experiences. Participants described a lack of a “therapy mindset” (I: 35-44, F) when not in the clinician’s office but being in their home sanctuary. In addition, video consults could be distracting and unhelpful when respondents could observe their own expressions and reactions on the screen; others found the telephone method helpful as it gave them some control over how they showed emotion and less feeling of being scrutinised.


*Sometimes we’ll have really good conversations, but then I’ll be like… is someone from my family… has anybody heard? You can’t let your guard down in the way that you normally would.*
(I: 35-44, F)

#### 3.3.3. Clinician–Patient Relationships

People who sought treatment for mental health and other issues noted that caring and reassuring health professionals helped calm anxieties in this time of heightened stress and fear. They described having felt heard and supported. Having a pre-existing trusting relationship with a health professional facilitated those with mental distress to seek help.


*When I’m starting to hear voices or… getting intrusive flashbacks or intrusive thinking, what potential risk that poses in terms of talking to a health professional… I was really grateful that… I had an established relationship with my GP... And she followed up with me.*
(I: 45-54, F)

Participants described how care delivered over the phone worked only because of these established relationships and would likely not work otherwise. The following participant believed that were it not for the long-term relationship with their GP, they would have suffered in silence.


*I found it hard for the consult surrounding my mental distress to be upset on the phone. It felt less personal. I am so grateful that I had a longer-term relationship with my GP. If I hadn’t I would not have felt okay to contact a GP. I just would not have done it.*
(S: 35-44, F)

On the other hand, where relationships were not strong, this could lead to heightened anxiety.


*I had an absolutely terrible experience with (a call centre) when I was concerned I had COVID-19. The member of staff I spoke to was incredibly dispassionate, uninterested in helping me and useless, leaving me feeling anxious and helpless.*
(S: 18-24, F)

In turn, highlighting the importance of considering health needs beyond COVID-19, one participant suggested that service needs of some groups, such as those with a disability may not be met in a pandemic environment. Anxiety was exacerbated for this participant.


*I’m terrified as a disabled person that my health might be compromised during the pandemic, not just the risk of COVID-19 but that my health as a disabled patient isn’t a priority (to others).*
(S: 35-44, gender diverse)

### 3.4. Social and Economic Foundations

People reported a range of wellbeing effects related to employment and income, depending on how the lockdown affected their work and employment status. For some who continued to work (including those working from home or essential workers), the pandemic meant greater work pressures and longer hours. Participants described these situations as “psychologically demanding” (S: 55-64, F) and draining. These situations were exacerbated when employers placed additional expectations on employees, and by the tensions between work demands and household responsibilities (see Section 3.1.1).


*[My wife’s] boss was rather inflexible and wanted her staff to all do online meetings and professional development… which was equivalent to sort of full-time work… [This] made it quite challenging with a one-and-a-half-year-old who needs constant attention.*
(I: 25-34, M)

Others who were working at home had a different experience, especially if they had supportive employers and were constitutionally suited to working from home (i.e., those describing themselves as introverts). These people were more likely to benefit from having more time for exercise and self-care, particularly if they escaped a long commute to work (see Section 3.2.1). Individuals described feeling less stressed and having less need to interact with the outside world.


*I actually find it slightly less stressful because I have plenty of quiet time to myself… In my office, we’ve talked about that we will be more flexible… I’m likely to establish a regular routine where I work from home… I don’t really need to go to the office… it’s nice to work from home.*
(I: 45-54, M)

In contrast, some participants who effectively lost their employment because of lockdown restrictions on non-essential positions found difficulty coping. They described feeling “useless” (I: 55-64, M) if they were not earning an income.


*It was an awful letter they [the workplace] sent to everyone just saying you are an essential worker, you must come to work. If you are over 70 or a vulnerable person you may use your holiday leave and then take leave without pay... And there’s people that have to make those awful decisions about working for a place where they feel unsupported or not coping financially.*
(I: 45-54, F)

Many respondents reported financial stress and worries about actual or possible job loss due to the pandemic. Participants highlighted that there were potential long-term financial ramifications from job losses and economic instabilities. These stressors were mitigated to some extent by Government benefits and the ability of people to use annual leave entitlements but exacerbated by concerns over the viability of individual businesses.


*From going from a good pay to a little pay and still have to pay child support, I had to use a bit of my annual leave, just to survive this… I’m behind in all my bills, letters coming asking to pay.*
(S: 35-44, F)

People further reflected that, despite additional costs associated with working from home, there were still others in a worse position.


*We had some staff who said,… ‘It’s going to cost us so much more for power and heating all day while we’re working’. It was like, but you are working, you are getting an income,… there are people who have lost their lives,… their jobs,… their livelihood.*
(I: 45-54, F)

## 4. Discussion

This study builds on a much needed and growing base of qualitative research into the experiences of people in managing the effects of lockdown during a pandemic. The richness of findings demonstrates multifaceted changes in how people worked and lived over this time. These changes were seen in four main areas:The role of the community, households, and long-distance relationships;The dichotomy between feeling safer at home and undergoing positive behaviour changes, and feeling stuck at home while managing work and health issues;Accessibility and appropriateness of formal mental health support; andSocial and economic factors.

Individuals juggled their household and work responsibilities, and changes in usual routines. Many described successes in this juggling act through reflection on changing personal values and enjoyment of time with families. Such findings provide greater context to research highlighting positive changes to mental health and wellbeing during the pandemic [5,51,59,60,69]. Notably, however, participants described non-COVID distresses that were exacerbated and income and employment uncertainties because of the lockdown. The impacts of COVID-19 (particularly amongst women), including balancing employment and family, mental health, and domestic violence, are increasingly documented [31,32,33,47,58,70,71,72]. Applying the recovery plan framework to the data analysis strongly shows that even where a country has established such a structure for supporting wellbeing, the general population still face challenges. These challenges are perhaps not well managed within Aotearoa New Zealand’s health and social support system, particularly given the high levels of mental health and wellbeing problems prior to the first lockdown in 2020 [73,74].

To cope with multiple lockdown stressors and changes to usual routines, participants cited pursuits they undertook, including exercise, talking with friends and family, and reflecting on their slower pace of life. Interestingly, faith-based support did not emerge as a comfort mechanism for our research participants. How long routine changes are sustained after lockdown is yet to be fully evaluated; however, at the time of the present study, those with positive behaviour changes appeared conscious about the need to maintain good habits. Changes in physical activity [75], diet [76], substance use and other health behaviours [77,78] are highlighted in the international literature, often suggesting negative implications because of distancing and COVID-19. Other literature suggests that home-based activities such as exercise, yoga, and relaxation [79,80] and access to green spaces [35,81] can improve wellbeing during the COVID-19 pandemic, although perhaps only for more mobile members of the population. It is imperative to increase awareness of the importance of ensuring that lockdowns do not stop access to green spaces or to regular, effective health promotion messages. This includes having active pro-health and exercise messaging for individuals through media (television, radio, internet) and via health providers.

For many participants, there was a need to continue to work while based at home. Self-described introverts coped better than extroverts and were more suited to this sort of working environment. Significantly, individuals highlighted the need for organisational support to facilitate a safe work-from-home environment, a finding supported by earlier literature [82]. One step in ensuring this safety is through encouraging the adoption of flexible work policies wherever possible to encourage worker autonomy and management of multiple work/family responsibilities, typically for women and carers with young children. These policies should include consideration of necessary office equipment, ergonomic set-ups, agreed hours of work, and reporting expectations.

Home as a place of safety was articulated by many participants; however, for some, contact with those outside their physical location was needed to maintain wellbeing. Such findings align with earlier work describing the nourishing nature of a spiritual home for Māori outside of that available solely from a physical place, and the importance of interconnectedness through online means to facilitate wellbeing over lockdown [83,84]. The increasing reliance on telecommunications, particularly videoconferencing [85], enabled contact with work, but also with family, friends, and social/health services. Using videoconferencing was a new skill for many with the need to upskill and support populations at risk of digital exclusion, [86,87] particularly older people and those at risk of isolation. This work also highlights the risk that those who cannot afford telecommunications services miss out on vital external connections.

Lockdown exacerbated mental distress for some. Services were not always responsive, and people were not necessarily engaged with accessing telephone/videoconferencing services, or able to access informal community support. Our earlier work suggests that individuals delayed seeking health care during lockdown [88]. Those with pre-existing mental health issues fared better when they had ready access to primary care or other services, especially if they knew the clinician. Some mentioned a sense of solidarity as improving their ability to cope under lockdown. Reinforcing feelings of solidarity in light of a loss of space and social relations may offer a framework to start grounding practical recovery steps [89]. Our findings suggest two main paths to improve mental health service provision during pandemics. The first relates to ensuring there is greater clarity and consistency in the public health messaging about access to mental health and wellbeing services, and that these services are agile to service user needs including provision of emergency services. Secondly, there remains a need for clinicians to: provide choice to service users around accessing telehealth, be aware of pros and cons of using phone or videoconferencing, seek service user preference, and specifically monitor changes in clinical outcome, which may relate to mode of care delivery [90,91].

Our study showed that lockdown restrictions caused a loss of autonomy and, for some, this impacted mental wellbeing. In future, plans are needed to safely enable social interaction without increasing the risk of future waves of transmission. In Aotearoa New Zealand, the first step has involved public buy-in, with vaccination efforts focussed on bubble expansion in vaccinated populations thus enabling family autonomy to interact with and care for loved ones. Other possible high-level ways to mitigate the impact of lockdown or similar severe restrictions on distancing on wellbeing are outlined in Table 4. These recommendations aim to improve solidarity and operate at five levels: clinical practice, working from home, healthy lifestyles, minimising digital exclusion, and integrated responses.

### Limitations and Future Research

This paper provides a snapshot of the impact of a COVID-19 lockdown on a segment of Aotearoa New Zealand’s population. With regard to limitations, it is not possible to distinguish the impact of the COVID-19 pandemic independent of the impact of a lockdown without the pandemic on participants. Owing to the snowball sampling method used in recruiting survey participants, which began with university promotion of an online survey, this group is likely to be reflective of a population less impacted by the economic effects of lockdown. As such, this research is limited in its ability to reflect the views of the most disadvantaged and struggling populations.

This work is also an analysis of research conducted in real-time by researchers also experiencing lockdown. Interview participants spoke spontaneously and in depth about deeply private experiences. The use of telecommunications tools supported this information sharing and participants said they appreciated having the opportunity to talk to an interviewer who was external to their bubble. Notably, the first 2020 lockdown occurred over the autumn months in Aotearoa New Zealand, with mild weather and less need for home heating, meaning that people’s views may differ had there been worse weather.

Future research into people’s wellbeing over the 2021 (and any subsequent) lockdowns would be timely, particularly given the politicisation of vaccine hesitancy and societal divisions based on vaccine status [92], and the waning of ‘be kind’ messaging. Such work could focus on how governments, community and social agencies, and health providers could ideally respond to empower individuals, families, and neighbourhoods working through the pandemic’s repercussions given their experiences over 2020. Given the impact of COVID-19 lockdown on younger populations [93], it would also be worthwhile to separately explore changes to their wellbeing as part of future research.

## 5. Conclusions

Our study relates how individuals and families reported on their wellbeing during a COVID-19 lockdown. Lockdowns will invariably have a legacy effect on populations, and findings from this research are particularly pertinent given how the pandemic has changed formal and family support structures. Where individuals were traditionally outwardly focused when looking for work comradery, social support, and health care, in our participant population, a change towards inward focussing behaviours occurred. Interventions that reinforce connectedness, continuity, and provide certainty may function as helpful directions for those facing social isolation. Such interventions include those that support local community connectedness, reduce digital exclusion, promote person-centered care, and support individuals effectively working from home. This is vital to ensure ‘recovery’ given the pandemic’s serious ongoing economic and health-related burdens.

## Figures and Tables

**Figure 1 ijerph-19-02269-f001:**
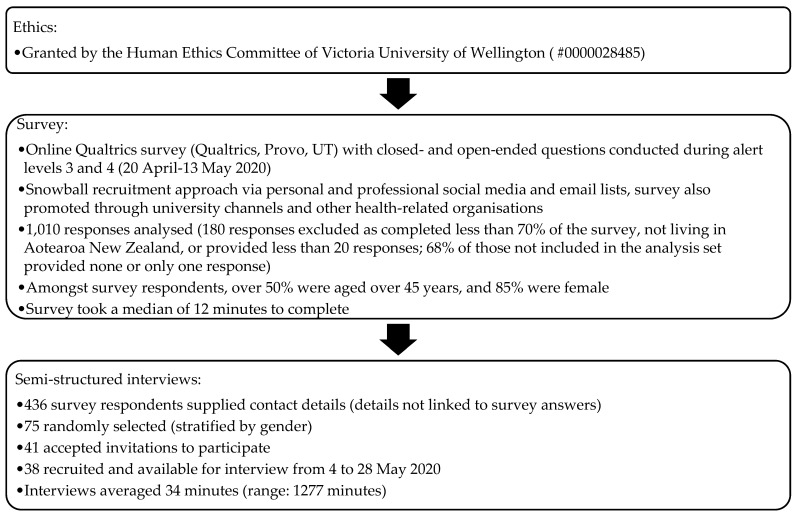
Data collection.

**Table 1 ijerph-19-02269-t001:** Recovery framework areas of impact.

Focus Area	Desired Outcome
Social and economic foundations for psychosocial and mental wellbeing	Whānau and communities have the resources and supportive environments on which psychosocial and mental wellbeing is built.
Community-led solutions	Whānau and communities are empowered and supported to respond to mental distress and lead recovery solutions.
Whānau and individuals look after their mental wellbeing	People know how to look after their mental wellbeing and know where to get help if they need it.
Primary mental health and addiction support	Whānau and communities have free and easy access to mental wellbeing support services in their communities.
Specialist services	People with severe mental distress and addictions and their whānau get high quality timely mental health and addiction support.

**Table 2 ijerph-19-02269-t002:** Interviewee characteristics.

Age Range (Years)	Interviewees (*n* (%))
18–34	7 (18)
35–44	6 (16)
45–54	12 (32)
55–64	3 (8)
65+	10 (26)
**Gender ***	
Female	24 (63)
Male	14 (37)
**Prioritised ethnicity**	
Māori	6 (16)
Pacific peoples	3 (8)
Asian	4 (11)
New Zealand European/Other	25 (66)
**Current work status**	
In paid employment without change caused by COVID-19	22 (58)
In paid employment with reduced pay due to COVID-19	3 (8)
Not in paid employment and not looking for a job	13 (34)
**Grouped District Health Board (DHB) region ^#^**	
Northern region	7 (18)
Midland region	3 (8)
Central region	20 (53)
South Island	8 (21)

* Interviewees did not identify as “gender diverse”, survey participants who described themselves as “gender diverse” or “prefer not to say” were grouped as these were small numbers; ^#^ Northern region = Northland, Waitematā, Auckland, and Counties Manukau DHBs; Midland region = Waikato, Bay of Plenty, Tairāwhiti, Lakes, and Taranaki DHBs; Central region = Whanganui, Hawke’s Bay, MidCentral, Wairarapa, Hutt, and Capital and Coast DHBs; South Island = Nelson-Marlborough, West Coast, Canterbury, South Canterbury, and Southern DHBs.

**Table 3 ijerph-19-02269-t003:** Subthemes arising within the recovery framework.

Focus Area	Subthemes		
Community-led solutions	Household relationships and responsibilities	Telecommunications	Community and neighbourly relationships
Whānau (family) and individuals look after their mental wellbeing	Physical lifestyle and behaviour changes	Pace of life and mindset	Stuck or safe at home
Primary and specialist mental health and wellbeing support	Accessibility	The form of service delivery	Clinician-patient relationships
Social and economic foundations			

**Table 4 ijerph-19-02269-t004:** Recommendations to mitigate wellbeing burden.

Recommendation	Who?	Description
Opportunities for person-centred care	Health professionals, funders	Recognise the diverse needs of populations and high-risk groups whose wellbeing may be greatly affected by lockdown Create COVID-19 specific lived experiences groups Provide proactive mental health support for communities, families, and individuals, including a recognition of the need to continue this outside of lockdown with a focus on hard-to-reach communities Proactively offer increased health and social care services via phone or videoconferencing, particularly for those with disabilities and time-sensitive care needs, including the terminally ill Offer technical support and advice for those unfamiliar with telehealth approaches
Support for working from home options	Employers	Flexible work hours, with particular attention paid to the needs of families with pre-schoolers and parents trying to home school primary school-aged children Support to set up and manage home offices, including perhaps stipends for internet or office equipment Job retention schemes Work sharing schemes Updating workplace wellbeing policies
Promote positive coping strategies and messaging	Central and local government, health professionals	Set up pathways to support creation of greener communities (including access to nature and community gardens) Support for businesses setting up and individuals conducting online shopping with deliveries for essential items Establish a national knowledge repository of COVID-19-specific wellbeing resources and interventions Encourage the de-stigmatisation of discussions around mental health Raise awareness about the difference between mental illness and mental distress Encourage community/neighbourhood connectedness through community digital communications Ensure those at risk of domestic violence/family harm can get away from their situation and know how to access help
Reduce digital exclusion	Central government, tertiary education providers, health professional representative organisations	Mitigate the effect of population inequities through providing routes to low-cost digital solutions Consider funding telecommunications services for older populations or those in recognised need Create specific telehealth clinical training pathways so that service delivery meets patient needs
Integrated whole of society response to COVID-19	Workplaces, government and social services	Build stronger communities through fostering links between volunteering organisations, and other non-government organisations Maintain focus on public health messaging, including disease prevention strategies focussing on alcohol, tobacco, and gambling Engage in providing lifestyle programming with a focus on health and wellbeing, learning new skills, and accessing financial support Create primary and secondary school programmes aimed at encouraging mindfulness and providing life skills. Engage in discussions promoting full vaccine uptake Include Māori and marginalised communities in policymaking to facilitate culturally safe responses to COVID-19

## Data Availability

Not applicable.

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
