# Peer review of "COVID-19 Pandemic Lockdown and Wellbeing: Experiences from Aotearoa New Zealand in 2020"

_ijerph, 2022, doi:10.3390/ijerph19042269_

Round 1
Reviewer 1 Report
Thank you for the opportunity to review this study entitled “COVID-19 pandemic lockdown and wellbeing: Experiences from Aotearoa New Zealand in 2020” (ijerph-1599285).
The manuscript presented an investigation about the explore environmental and social factors that influenced the wellbeing of individuals during the first lockdown in Aotearoa New Zealand. Participants were 624 individuals (52.5% females). A six-factor structure and good model fit emerged from the analyses.
In my opinion the research topic is relevant, and the study is interesting. However, there are some minor issues that need to be addressed before the paper will be suitable for publication.
- The abstract is well written. Just a little note: the information about the samples (both the 1010 subjects who responded to the survey and the subset of surveyed individuals involved for the semi-structured interviews) should be deepened (Mean age and SD? Percentage of men and women?) to provide a clear picture of what will be presented in the paper.
- In my opinion, it would be good to refer to trend or longitudinal studies, if any. Since the authors frame this study considering the impact that COVID-19 has on a psychological level on people, I suggest some research to broad this aspect inthe introduction, which should be supplemented with further literature search by the authors:
-Hyland et al., 2021; doi: 10.1016/j.psychres.2021.113905.
-Gori & Topino, 2021; doi: 10.3390/ijerph18115651
-Wang et al., 2020; doi: 10.1016/j.bbi.2020.04.028
To find the suggested articles, the authors can use this source: https://www.doi.org/
- The authors declared to use of a mixed-methods approach and they wrote about an "anonymous online survey" hosted on Qualtrics. This at first made me assume that there were both qualitative and quantitative data. Instead, I understand that the online survey also contained open questions. I would like to be clearer in this regard, in this section and consequently in the abstract.
- Lines 692-710 seem to deal with the limitations and suggestions for future research. However, this is not immediate to the reader, singe the authors pass from the discussion of the results to these aspects without clarifying that we are moving on to exploring the limits. To promote the legibility of the paper, I recommend adding a sentence that makes it explicit (e.g., This research presents some limitations that should be addressed ") or a sub-paragraph.
- I suggest expanding the concluding section by deepening the applicative implications of this interesting research.
Reviewer 2 Report
This paper conducted a qualitative study to explore the environmental and social factors that influenced the wellbeing of individuals during the COVID-19 pandemic lockdown in Aotearoa New Zealand, presenting a significant problem of the COVID-19 pandemic. With some minor amendments it will be a suitable paper to publish.
(1) On page 5. "Data were first analysed by FI and LM using NVivo 12 (QSR International Pty Ltd) to manage the data and iteratively reviewed, checked, and interpreted by EM and KMH." I think it's inappropriate to use the authors' abbreviations here, because it may make the reader think they are abbreviations for some terms.
(2) I suggest the authors briefly describe why thematic analysis was chosen over other methods, such as grounded theory analysis.
If these issues can be addressed the paper deserves publication.
